# Inhalation Bioaccessibility and Risk Assessment of Metals in PM_2.5_ Based on a Multiple-Path Particle Dosimetry Model in the Smelting District of Northeast China

**DOI:** 10.3390/ijerph19158915

**Published:** 2022-07-22

**Authors:** Siyu Sun, Na Zheng, Sujing Wang, Yunyang Li, Shengnan Hou, Qirui An, Changcheng Chen, Xiaoqian Li, Yining Ji, Pengyang Li

**Affiliations:** Key Laboratory of Groundwater Resources and Environment, Ministry of Education, College of New Energy and Environment, Jilin University, 2519 Jiefang Road, Changchun 130021, China; ssy19@mails.jlu.edu.cn (S.S.); sjwang20@mails.jlu.edu.cn (S.W.); liyunyang@iga.ac.cn (Y.L.); houshengnan@iga.ac.cn (S.H.); anqr21@mails.jlu.edu.cn (Q.A.); chencc21@mails.jlu.edu.cn (C.C.); xqli20@mails.jlu.edu.cn (X.L.); jiyn20@mails.jlu.edu.cn (Y.J.); pyli19@mails.jlu.edu.cn (P.L.)

**Keywords:** bioaccessibility, deposition fraction, exposure assessment, PM_2.5_

## Abstract

PM_2.5_ can deposit and partially dissolve in the pulmonary region. In order to be consistent with the reality of the pulmonary region and avoid overestimating the inhalation human health risk, the bioaccessibility of PM_2.5_ heavy metals and the deposition fraction (DF) urgently needs to be considered. This paper simulates the bioaccessibility of PM_2.5_ heavy metals in acidic intracellular and neutral extracellular deposition environments by simulating lung fluid. The multipath particle dosimetry model was used to simulate DF of PM_2.5_. According to the exposure assessment method of the U.S. Environmental Protection Agency, the inhalation exposure dose threshold was calculated, and the human health risk with different inhalation exposure doses was compared. The bioaccessibility of heavy metals is 12.1–36.2%. The total DF of PM_2.5_ in adults was higher than that in children, and children were higher than adults in the pulmonary region, and gradually decreased with age. The inhalation exposure dose threshold is 0.04–14.2 mg·kg^−1^·day^−1^ for the non-carcinogenic exposure dose and 0.007–0.043 mg·kg^−1^·day^−1^ for the carcinogenic exposure dose. Cd and Pb in PM_2.5_ in the study area have a non-carcinogenic risk to human health (hazard index < 1), and Cd has no or a potential carcinogenic risk to human health. A revised inhalation health risk assessment may avoid overestimation.

## 1. Introduction

PM_2.5_ is particulate matter with an aerodynamic diameter of less than 2.5 μm. Epidemiological studies show that if the average concentration of PM_2.5_ increases by 10 μg·m^−3^ every two days, the daily cardiovascular mortality rate will increase by 0.36–1.22%, and the daily respiratory mortality increases by 0.74–1.78% [1,2,3]. If the weekly concentration of PM_2.5_ increases by 10 μg·m^−3^, the weekly total lung cancer mortality rate will increase by 6.2% [4]. PM_2.5_ concentration is related to severe obstructive and chronic diseases of the respiratory and cardiovascular systems such as asthma and emphysema [5,6]. The metal in the composition of PM_2.5_ only accounts for 2–8%; however, due to the long half-life, it may be hard to be excreted when it enters the human body, and then it will be accumulated in the human body [7,8,9]. Heavy metals could enhance the oxidative stress response of lung epithelial cells, produce enzyme activity, affect the cell cycle, trigger cell apoptosis, and cause lung function injuries [10,11,12,13]. Cell exposure experiments prove that exposure to low concentrations of heavy metals can also affect normal cellular immune responses [14]. Mixed metals are more toxic to cells than single ones. The apoptosis rate of A549 cells after exposure to mixed metals could reach 36.5% [13,14,15,16]. The results of toxicity experiments on lung cancer cell lines and rats show that heavy metals in particles can inhibit the expression of proteins related to redox homeostasis in cells, and overexpresses the proteins that are related to DNA damage [17,18]. It is significant to study the PM_2.5_ heavy metals in the lungs for human health.

When PM_2.5_ enters the respiratory system, it will deposit in the head (above the pharynx), the trachea bronchus, and the pulmonary region [19,20,21]. The results of exposure experiments validate that the smaller the size of PM, the easier it is to deposit in the pulmonary region. Only 30% of the gold nanoparticles are deposited in the tracheal epithelium, which may quickly excrete with the movement of cilia. PM deposited in the pulmonary region may stay for one to several days [19,20,21,22,23,24,25]. The studies proved that the metal nanoparticles deposited in the pulmonary region were still partially undissolved after 24 h, and the water-soluble metal ions could be cleared faster in the pulmonary region. The rate at which metal deposited in the pulmonary region releases into the systemic circulation is related to its solubility [23,24,26]. Only the components of PM_2.5_ that can dissolve in the pulmonary region can participate in human circulation and affect human health [27,28,29]. The previous studies, which are focused on total concentration of PM heavy metals, may overestimate the human health risk.

The mice exposure results may be more similar to those in humans, but the cost is higher [30,31,32]. Many studies used more economical and convenient in vitro simulated lung fluid to simulate the dissolved concentration of PM in the lungs [27,28,29,33,34]. There are two types of in vitro simulated lung fluid commonly used in research: one is the artificial lysosomal fluid (ALF, pH = 4.5 ± 0.1) which simulates the acidic macrophage deposition environment; the other is the Gamble’s solution (GS, pH = 7.4 ± 0.2) which presents the neutral depositional environment of the lung interstitium [34,35,36]. The lung is composed of acidic macrophages and neutral lung interstitium [37]. Therefore, the inhalation exposure concentration of PM for a human may be between those two. The bioaccessibility of simulated lung fluid for human health risk assessment is widely used. Studies suggested that the bioaccessibility concentration of metals should be considered in the health risk assessments [28,38,39,40]. The multiple-path particle dosimetry (MPPD) model is used to estimate the PM deposition fraction (DF) in the respiratory system [19,21,41]. The total concentration of heavy metals and the sampling mass of PM_2.5_ are taken into consideration in health risk. There may be differences with the reality of the pulmonary region, and the health risks of PM_2.5_ of heavy metals may be overestimated; therefore, there is an urgent need to revise the inhalation health risk of PM_2.5_ using bioaccessibility of heavy metals and deposition fraction.

This study takes Huludao City as the research area to discuss and analyze the bioaccessibility of Cd and Pb in PM_2.5_, the DF of PM_2.5_, and the inhalation human health risk assessment based on the above two factors. Additionally, the Monte Carlo method is used to analyze the probability. The objectives of this study are as follows: (1) determine the bioaccessibility of Cd and Pb in PM_2.5_ in the study area; (2) determine the particle DF in the respiratory system for different age groups; (3) determine the inhalation exposure doses for different populations; and (4) compare the evaluation results of different inhalation exposure doses and their differences and analyze their probability distribution.

## 2. Materials and Methods

### 2.1. Study Area

The study area, Huludao City, is located in Liaoning Province in northeast China, near the Liaodong Gulf. The average annual rainfall is 590 mm, and the primary wind direction in summer is southwest and in winter it is north. Huludao Zinc Plant (HZP) is the largest zinc smelting plant in Asia, situated at the southeast of Huludao city, and 97.3% of Cd and 89.6% of Pb discharged in the waste gas of Liaoning Province were from HZP, respectively (2003) [42]. Jinxi Chemical Plant (JCP) is another major industrial enterprise in Huludao City [43]. Many metals are emitted into the environment in large quantities through atmospheric deposition and solid waste emissions [43].

### 2.2. Sampling Collection and Preparation

The distribution of sampling sites is shown in Figure 1. At the Xinqu Park (XP) and Dongcheng District (DD), the PM_2.5_ mass concentrations and the atmospheric particulate samples were collected separately by using two APM systems (i-5030 and i-FH62C14, Thermo Fisher, Franklin, MA, USA). The duration of both sample collections was 6 h and the airflow was 16.7 L·min^−1^. The PM_2.5_ was sampled in the study areas from January to December 2015 at two sites, and a total of 40 valid samples were collected. Due to the life, work, and transportation of residents, they will move within the area. Most of the urban residents, commercial centers, and hospitals are located in the sampling sites [44].

The PM_2.5_ samples were collected on 47 mm-diameter quartz fiber filters. Before sampling, the quartz filters were heated up to 550 °C in closed aluminum foil pouches for 6 h, to decrease their metal contents [45,46]. After collecting the sample, it was stored at −20 °C until analysis.

### 2.3. Chemical Analyses

The filter remained at a constant temperature (25 °C) and relative humidity (30%) for 24 h, and was weighed using a high-precision electronic balance, then cut into quarters, and transferred to a Teflon digestion vessel [47,48,49]. Then, HNO_3_, HF, and HClO_4_ (1:1:5) were added to the vessel. The sample mixtures were heated for 3 h at 200 °C [50]. After digestion, the samples were cooled to room temperature, and dissolved in nitric acid solution, and the solution volumes were 10 mL, then filtered with a 0.22 μm polyethersulfone filter. The sample digests were analyzed using inductively coupled plasma mass spectrometry (ICP-MS) (Thermo Fisher, Bremen, Germany) to determine the concentrations of Cd and Pb on the filters [51]. For quality assurance/quality control, river sediment standards (fluvial sediments: GBW08301) were digested and analyzed as described above to determine heavy metal detection limits and recovery rates. The average concentrations in the blank filters were the following (μg·m^−3^): 0.007 ± 0.004 Cd; and 0.026 ± 0.016 Pb. The detection limits were the following (μg·m^−3^): 0.010 Cd and 0.039 Pb, this is consistent with the metal detection limits in previous studies [52,53]. According to the river sediment tests, the average recovery rates of heavy metals ranged from 86.24% to 115.97%, the metal recovery rate meets the requirements [54,55,56]. The other details are described in existing studies [48].

### 2.4. Bioaccessibility of Inhalation

Risk assessments associated with in vivo animal studies are time-consuming, expensive, and raise ethical concerns, while in vitro bioavailability assays are considered an alternative to in vivo measurements because they correlate well with in vivo bioavailability experiments for certain metals [30,39,40,57,58]. Due to the differences in the composition of different simulated lung fluids in vitro (such as inorganic salts, proteins and phospholipids, etc.), they may have different effects on the dissolution of heavy metals and the simulated lung fluid is selected for different research needs [27,28,59,60,61].

Two simulated lung fluids, the artificial lysosomal fluid (ALF, pH = 4.5 ± 0.1) and the Gamble’s solution (GS, pH = 7.4 ± 0.2), were used to measure Pb and Cd inhalation bioaccessibility in PM_2.5_ [34,40,62]. The chemical combinations of simulated lung fluid referred to the existing studies [34,62,63]. The alveolar surfactant 1,2-dipalmitoyl-sn-glycerol-3-phosphatidylcholine (DPPC) was added to the GS to simulate the lung fluid [27,61]. The solutions prepared in this study are analytical chemicals and deionized water. Additionally, to prevent the influence of bacteria and fungi on the experimental results during the simulation process, the deionized water was subjected to sterilization treatment (at 120 °C for 20 min) [62].

The bioaccessibility experiment in this study involved cutting a quarter of the filter, transferring it into a 50 mL polypropylene tube, and adding 30 mL of simulated lung fluid [38,61]. To simulate the deposition environment, the sample remained at 37 °C for 24 h in the dark, at 100 rpm for a day, then was centrifuged at 4000 rpm for 10 min. Next, 4ml of the supernatant was taken, 1 mL of 3% nitric acid solution added, was filtered with a 0.22 μm polyethersulfone filter, and stored at 4 °C until analysis [40,63]. The samples were analyzed using ICP-MS (Thermo Fisher, Germany) to determine the concentrations of Cd and Pb.

### 2.5. Multiple-Path Particle Dosimetry (MPPD) Model

The MPPD model (V3.04, ARA Inc., Arlington, VA, USA) is based on a stochastic lung airway morphometry model [22,41] that represents an asymmetric and more realistic geometry of the human lung to obtain more realistic deposition results. The MPPD model could provide the most realistic conditions for the study population in terms of anatomy and physiology and quantify the total deposition volume and regional deposition volume of each generation of airways, and could more clearly understand the deposition rate of particulate matter in the human respiratory tract. Compared with the in vivo model, it has higher accuracy and is more convenient, and has been widely used to simulate airway deposition [19,20,64]. However, factors such as exposure scenarios, physiological input parameters, age, and gender may lead to differences in deposition fraction; thus, it is necessary to consider lung models under different conditions and parameter selection [19,64,65].

In this study, the stochastic lung airway morphometry was adopted. According to the USEPA’s exposure parameter manual, the long-term exposure weight and respiratory parameters of different age groups were taken [66,67], taking the average of each age group for calculation. Therefore, in this study, the children were divided into 2–23 years old, and the adults were 23–96 years old. The functional residual capacity (FRC) of children and adults was calculated using a formula, which was related to the height of children, the age and height of adults [20,68]. The heights of different age groups were determined according to existing studies [69,70,71]. The breath frequency, upper respiratory tract volume, and tidal volume of the population were based on existing studies [20,70]. Particles were assumed to be spherical with a density of 1.0 g·cm^−3^ [19,22,64]. Among the nasal, oral, oronasal, and endotracheal breathing conditions available in the model, nasal breathing was chosen. All the particles were assumed to enter the respiratory tract through the nose [22]. The exposure environment is the upright body direction. Some existing research on sedimentation parameters of people of different ages are referred to in this research [20,68].

### 2.6. Exposure Risk Assessment

PM_2.5_ enters the body through the respiratory system, seriously damaging human health [72,73]. To determine the inhalation risk of PM_2.5_ caused by Cd and Pb in the study area, a health risk assessment was carried out according to the guidelines of the USEPA. Because of the influence of different exposure doses (bioavailable concentration, deposition concentration, and both factors) on the assessment results, risk assessment of different doses and probability analyses were carried out [21,28,33,65,74].

According to the Human Health Assessment Guidelines of the USEPA (2011), this study estimates the risk of inhalation of Pb and Cd in PM_2.5_. The following formula is used to calculate the average daily inhaled dose of non-carcinogenic metals (ADD_inh_, mg·kg^−1^·day^−1^) and the average daily inhaled dose of metal carcinogens exposed to during daily life (LADD_inh_, mg·kg^−1^·day^−1^) [28,75].
(1)$${\mathrm{ADD}}_{\mathrm{inh}\;i}\left({\mathrm{LADD}}_{\mathrm{inh}\;i}\right)={\;C}_{i}\times \mathrm{InhR}\times \mathrm{EF}\times \mathrm{ED}/(\mathrm{BW}\times {\mathrm{AT}}_{n}\left({\mathrm{AT}}_{c}\right))$$

C_i_: different assessment concentrations of heavy metals, mg·m^−3^; InhR: inhalation rate, m^3^·day^−1^; EF: exposure frequency, 350 day·year^−1^ in this study; ED: exposure duration, 26 years for adults, and 6 years for children; BW: average body weight; AT_n_: averaging time for non-carcinogens, ED × 365 days; AT_c_: averaging time for carcinogens, 70 × 365 days; age of children 2–23 years group and adults 23–96 years group [66,67]. The calculated concentrations in this study are the bioavailable concentration in the simulated lung fluid (C_ALF_, C_GS_), the PM_2.5_ deposition concentration (C_DF_), and then considering the bioaccessibility and the DF (C_ALF+DF_, C_GS+DF_). The concentration of simulated lung fluid is measured through bioaccessibility experiments, and the calculation equations for other concentrations are shown in Equation (2), DF represents the deposition fraction of PM_2.5_ in the pulmonary region.
(2)$$C_{\mathrm{DF}}={\;C}_{\mathrm{Total}}\times \mathrm{DF}$$
(3)$$C_{\mathrm{ALF}+\mathrm{DF}}={\;C}_{\mathrm{ALF}}\times \mathrm{DF}$$
(4)$$C_{\mathrm{GS}+\mathrm{DF}}={\;C}_{\mathrm{GS}}\times \mathrm{DF}$$

The hazard quotient (HQ) represents the risk due to non-carcinogens, and the incremental lifetime cancer risk (ILCR) for carcinogens was calculated using the following equations [28,75]:(5)$${\mathrm{HQ}}_{i}={\;\mathrm{ADD}}_{\mathrm{inhi}}/{\mathrm{RfD}}_{i}$$
(6)$${\mathrm{ILCR}}_{i}={\;\mathrm{LADD}}_{\mathrm{inh}\;i}\times {\mathrm{CSF}}_{i}$$

RfD: reference dose for Cd 1.00 × 10^−3^ mg·kg^−1^·day^−1^; Pb is 3.50 × 10^−3^ mg·kg^−1^·day^−1^; CSF: carcinogens slope factor for Cd 6.30 (mg·kg^−1^·day^−1^)^−1^ [28,75].

Without considering the interaction between heavy metals, the hazard index (HI) was calculated by superimposing the HQ value of each heavy metal as shown in Equation (7) [28,75].
(7)$$\mathrm{HI}\;=\underset{i}{\sum }{\mathrm{HQ}}_{i}$$

### 2.7. Monte Carlo

For health risk assessments, the Monte Carlo simulation method was used to evaluate the uncertainty, this process was performed using the crystal ball software (version 2000.2, Decisioneering, Denver, CO, USA). All calculated cumulative probabilities are reported in 5000 simulation iterations [28,76,77]. The concentration and body weight are average values, and the respiratory rate is normally distributed.

## 3. Results and Discussion

### 3.1. Particulate Mass and Metal Concentrations

The mean concentrations of PM_2.5_ are 74.70 μg·m^−3^ in Xinqu Park (XP), 88.75 μg·m^−3^ in Dongcheng District (DD), and in Huludao City 81.72 μg·m^−3^, which reach 2.13, 2.54, and 2.33 times higher than the value (35.00 μg·m^−3^) recommended in the National Ambient Air Quality Standard of China (NAAQS, 3095-2012). The concentration of DD is higher than that of XP, which may be due to the different locations of the two sampling sites from the pollution source. A non-parametric test was performed on them, and the results showed that there were no significantly different (*p* > 0.05) concentrations at the two sites. The concentration of PM_2.5_ is similar to that of Beijing but lower than that of Zhuzhou, an industrial city [28,78]. This may be because the sampling site in Zhuzhou is located in the main industrial area, and although the sampling site in the study area is affected by industry, it is not near the emission source [78].

The mean concentrations of Cd and Pb are 9.99 and 263.9 ng·m^−3^ in XP, 11.39 ng·m^−3^ of Cd, 452.1 ng·m^−3^ of Pb in DD, and 10.86 and 358.0 ng·m^−3^ in all study areas. Additionally, the metal concentration of DD is higher than that of XP. The non-parametric test results of the two sites validated that there was no significant differentiation in the concentration of Pb (*p* > 0.05), but the concentration of Cd was diverse (*p* < 0.05). That may be due to the difference in the source of Cd between the two sites, the distance between DD and the HZP and JCP is closer, which may be more affected [43].

### 3.2. Bioaccessibility of Metals

The concentrations of PM_2.5_ metal in simulated lung fluid and bioaccessibilities are shown in Table 1. For Cd, the bioaccessible concentrations are 2.13–3.50 ng·m^−3^. This result is higher than Beijing and lower than Nanjing’s industrial area which is influenced by petrochemical and metallurgical industries [28,79]. Although the study area is affected by metal smelting, there is a distance between the sampling site and the smelter. The result is distinguishing from the change in the total metal concentration, which may be due to the different form derived from the different sources of the metal, so the dissolved concentration in the lung simulation fluid is different [43,80].

Studies divided the bioaccessibility of metals into four levels: very high (>50%); high (30–50%); intermediate (15–30%); and low (<15%) [28,81]. Cd has high bioaccessibility in the ALF, intermediate and low bioaccessibility in GS (12.1% for XP). The bioaccessibility of ALF is higher than that of GS. This result is consistent with other studies finding that the bioaccessibility of metals in the acidic solution is higher than that of the neutral solution. It means that for most metals, the content of dissolved metals in the phagocytosis is higher than that of the neutral interstitial fluid of the lung [28,81]. The bioaccessibilities of Cd in ALF and GS are both lower than that of existing studies (>70% and >46%) [28,81,82]. It may be due to the differences in the bioaccessibility of Cd compounds obtained from different environments in these studies. Water-insoluble Cd mainly comes from lead smelters, coal combustion, and non-ferrous metal production [83]. The high soluble Cd mainly comes from waste incineration emissions [84]. Therefore, comparing with other studies, the PM_2.5_ collected in this study may contain higher levels of insoluble Cd compounds.

For Pb, the bioaccessible concentrations are 57.96–116.7 ng·m^−3^. This result is higher than Beijing and lower than Nanjing’s industrial area [28,79]. The bioaccessibility of Pb is high and intermediate in the ALF, intermediate and low in GS (12.4% for XP). The bioaccessibility of ALF is higher than that of GS. The bioaccessibilities of Pb in ALF and GS in this study are lower than that of existing studies (>50% and 36%) [28,81,82]. Studies show that the metal forms and the properties of different compounds may affect their leaching concentrations in simulation solutions [80]. Comparing with other studies, the PM_2.5_ of Pb in this study may contain more incompatible compounds.

The different sources of metals in PM and the dissolution rate of the components in the in vitro simulating fluid are numerous. The more soluble the components, the higher the bioaccessibility [83,84]. The metal of PM_2.5_ in the study area may have fewer soluble components and more insoluble components.

### 3.3. Deposition Fractions

In this paper, the semi-empirical MPPD model (V3.04, ARA Inc.) was used to estimate the deposition pattern of PM_2.5_ in various regions of the human respiratory system (head (H), trachea bronchus (TB), and pulmonary (P)). This model allows the calculation of the particle deposition fraction (DF) under different airway morphologies [41].

Figure 2 shows DFs of PM_2.5_ in different organs for different age groups. The total DF (the sum of the head, trachea, bronchi, and pulmonary region), for the children group in this paper, the oldest age group has the highest DF (63.76–78.84%); for adults, there is no significant change, the elderly (65–96) have the highest DF. Additionally, males have a higher deposition fraction than females at all ages, which is consistent with the results of existing studies [21]. Studies validated that PM deposited in the pulmonary region will stay for a day or longer [20,21]. Therefore, this study mainly discusses the pulmonary region. The DF of the pulmonary region gradually decreases with age, and males have higher fractions than females, with the highest in the 2–5-year-old male children, and the lowest in the 65–96-year-old female adults. Compared with the deposition parameters, this may be affected by FRC, breathing frequency (BF), and tidal volume (VT). Studies show that children are more susceptible to respiratory diseases than adults, and the elderly are the most susceptible [19,20].

### 3.4. Average Daily Inhaled Dose

Table 2 shows the average daily inhaled doses of Cd and Pb in PM_2.5_ for different age groups. Both Cd and Pb of PM_2.5_ deposit in the pulmonary region and gradually dissolve. In this study, the ADD_inh_ and LADD_inh_ of Cd and Pb were calculated using the bioaccessibility and the DF, which were based on Equations (1), (3) and (4), and related parameters. For ADD_inh_, the dose for males is higher than that for females, and this result is consistent across all age groups of children and adults in both Cd and Pb, which may be because the FRC, BF, inhalation, and other parameters of males are higher than those of females [21,67,85]. The dose for children is higher than that for adults, and similar results were proved in different assessments [19,20,70]. The dose of Pb is higher than that of Cd because the bioaccessible concentration of Pb is higher than that of Cd. For LADD_inh_, the dose for males is higher than that for females, and this result is consistent across all age groups of children, and in of both children and adults it was shown that LADD_inh_ gradually decreased with age. The LADD_inh_ in the 23–30-year-old group is the largest, which may be because of the longer exposure duration of the groups and the higher PM_2.5_ deposition dose than other age groups [21,67].

For the pulmonary region, it is composed of an acidic macrophage and a neutral lung interstitium [28,86]. Therefore, ADD_inh_ and LADD_inh_ may be the threshold, which may be between doses in the acidic intracellular deposition environment (ALF) and the neutral extracellular deposition environment (GS). Studies demonstrate that the PM will enter the lung interstitium by macrophages and return to the lung epithelium or other organs [23,24,26]. Within the study area, Cd doses for ADD_inh_ were 0.04–0.42 mg·kg^−1^·day^−1^ for females and 0.05–0.43 mg·kg^−1^·day^−1^ for males; Pb doses were 1.13–14.0 mg·kg^−1^·day^−1^ for females and 1.23–14.2 mg·kg^−1^·day^−1^ for males; and LADD_inh_ doses were 0.007–0.043 mg·kg^−1^·day^−1^ for females and 0.008–0.043 for males.

### 3.5. Human Health Risk Assessment

In this study, the rank of the HI value of the different inhalation exposure doses is ALF > DF (GS) > ALF + DF > GS + DF, where the HI values of DF and GS may be affected by the age and the human parameters (such as respiration rate and the metal chemical composition of PM_2.5_). The human health risks were evaluated by the weight, inhalation rate, DF, and the bioaccessible concentration of metals in different simulated lung fluids. Figure 3 shows the HI values. When the HI exceeds 1, the metal poses a non-carcinogenic risk to the human; when the HI is less than 1, it does not pose a non-carcinogenic risk to the human [87,88]. In the study area, metals do not pose a non-carcinogenic risk to people of different ages (HI < 1). The non-carcinogenic risk of children is higher than that of adults, the same as the study in Agra, northern India, and Zhuzhou, China [51,78]. Additionally, for different age groups, the non-carcinogenic risk of the population decreases with age. Children (2–5 years old) have the highest non-carcinogenic risk (0.014 for DF_male_, 0.018 for ALF_male_, 0.009 for GS_male_, 0.004 for ALF + DF_male_, 0.002 for GS + DF_male_), and adults have lower non-carcinogenic risk. The results of the different age groups demonstrate that the male HI is higher than that of the female. It may be due to children’s low weight and low inhalation [76]. Figure 3 shows that the assessment results that only consider the PM_2.5_ deposition fraction or bioaccessibility may be overestimated. This is consistent with the results of the mouse exposure experiment where only part of the PM deposits into the lungs. After 24 h or several days, metals with water solubility can appear in the extrapulmonary organs and circulate [23,24,26].

Figure 3 shows the ILCR values for different ages groups for different exposure doses. If the ILCR exceeds 10^−4^, the metal will have a carcinogenic risk to the human; if the ILCR is less than 10^−4^, exceeding 10^−6^, it will have a potential carcinogenic risk; if the ILCR is less than 10^−6^, it will not have a carcinogenic risk [87,88]. When only the bioaccessibility or DF of PM_2.5_ is considered, Cd will have a potential carcinogenic risk for adults and a carcinogenic risk to children; when both bioaccessibility and DF are considered, Cd is not carcinogenic to the human. The ILCR value of adults is higher than that of children, and males are higher than females, which is the same result as the study in Agra, northern India, and Zhuzhou, China [89,90]. For adults and children, the results demonstrate that the cancer risk for the population decreases with age. Adults (23–30 years old) have the highest risk of carcinogenesis (8.21 × 10^−7^ for DF_male_, 1.94 × 10^−6^ for ALF_male_, 1.18 × 10^−6^ for GS_male_, 2.69 × 10^−7^ for ALF + DF_male_, 1.63 × 10^−7^ for GS + DF_male_), and children (11–23 years old) have a lower risk of cancer.

Comparing the human carcinogenic risk and non-carcinogenic risk of different inhalation exposure doses, the results of this study suggest that the previous health risks may have been overestimated, and the revised assessment results may be more applicable to the reality situation.

### 3.6. Comparison of Differences in Human Health Risk Assessment

This study compares the inhalation human health risk assessment considering the PM_2.5_ deposition fractions, metal bioaccessibility, and both factors. A non-parametric test was performed on the results. There was a significant difference between the results of only considering the DF or bioaccessibility of metals and that considered both of the two (*p* < 0.05). The results of each assessment are different and may not be substituted for each other. Correlation analysis was undertaken on the evaluation results, and there was a significant correlation between the results. The use of the bioaccessibility of simulated lung fluid for human health risk assessment has been widely used [28,38,40]. The MPPD model is also used in human health risk assessment [7,91,92]. The comprehensive assessment results in this paper are related to the bioaccessibility and the PM_2.5_ deposition assessment results, indicating that the assessment results may also be reasonable.

This paper used Monte Carlo simulation to estimate the non-cancer risk and carcinogenic risk probability of different exposure doses. The cumulative probability of the HI values is shown in Figure 4. Each result is less than 1, indicating that the Cd and Pb in PM_2.5_ in the study area have a non-carcinogenic risk. The cumulative probability of the ILCR value is shown in Figure 4. The potential carcinogenic risk probability of ALF is 7.00–35.10% for children and 86.10–99.92% for adults; for GS it is 0.00–2.90% for children and 21.14–86.02% for adults; and for DF it is 0.00–10.72% for children and 0.02–52.16% for adults. The ALF + DF and GS + DF have no potential carcinogenic risks for humans in the study area. Considering only the bioaccessibility of metals or the DF of PM_2.5_ may cause overestimation. Exposure experiments showed that only part of the PM_2.5_ remains in the pulmonary region for more than 24 h. Moreover, some studies proved that the simulation results of in vitro bioavailability have a good connection with the results in mice [23,24,31,32,93]. Both bioaccessibility and DF are influential factors for human health risk assessment and the assessment results may be similar to the lung deposition environment and avoid overestimation.

This study only considered the deposition fraction of PM_2.5_ in the pulmonary region, and considered less the possible effects of other particle sizes on the deposition fraction of PM_2.5_; in addition, there are differences in composition and the bioaccessibility of heavy metals between simulated lung fluid and bronchoalveolar lavage fluid [24,32].

## 4. Conclusions

To avoid overestimating the human health risk of PM_2.5_ metal, this paper took Huludao City as the study area, used simulated lung fluid to simulate the PM_2.5_ metal bioavailable concentration and bioavailability in ALF and GS, and the MPPD model was used to simulate PM_2.5_ deposition in different age groups, then inhalation human health risk assessment was undertaken. The results proved that the Cd and Pb of PM_2.5_ in the study area had high and medium bioaccessibility (32.6–32.8% for ALF; 16.2–19.9% for GS); it was lower than the bioaccessibility of existing studies which may be due to more insoluble metal components. Additionally, the total DF in adults was higher than that in children in different age groups (0.64–0.79 for children; 0.72–0.80 for adults), while DF gradually decreased with age (0.15–0.26 for children; 0.12–0.14 for adults) in the pulmonary region. In addition, children were higher than adults, and males were higher than females. The range of the average daily inhaled dose should be between the acidic and neutral depositional environments of PM_2.5_ in the pulmonary region. Moreover, the dose for males is higher than that for females. The ADD_inh_ gradually decreased with the age of the groups (for Cd, 0.04–0.42 mg·kg^−1^·day^−1^ for females, 0.05–0.43 mg·kg^−1^·day^−1^ for males; for Pb, 1.13–14.0 mg·kg^−1^·day^−1^ for females, 1.23–14.2 mg·kg^−1^·day^−1^ for males), and the LADD_inh_ is higher for adults than for children (0.007–0.043 mg·kg^−1^·day^−1^ for females, 0.008–0.043 for males). The results of the human health risk and probability analysis of the study area with bioaccessibility (ALF, GS), deposition fraction (DF), combined bioaccessibility, and deposition fraction (ALF + DF, GS + DF) demonstrated that Cd and Pb in PM_2.5_ have no carcinogenic risk to human health, and Cd does not have any carcinogenic risk or potential carcinogenic risk. Thus, taking into consideration the bioaccessibility and deposition fraction of study, the simulated condition may be more similar to the pulmonary deposition environment, and could avoid the overestimation of inhalation human health risk.

## Figures and Tables

**Figure 1 ijerph-19-08915-f001:**
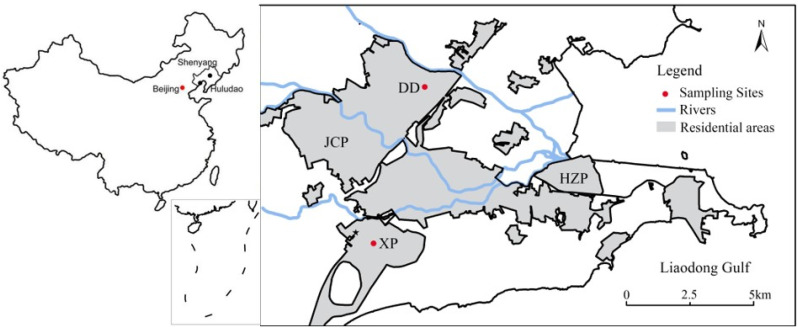
Study area and sampling sites. XP represents Xinqu Park sampling site; DD represents Dongcheng District sampling site; HZP represents Huludao Zinc Plant; JCP represents Jinxi Chemical Plant.

**Figure 2 ijerph-19-08915-f002:**
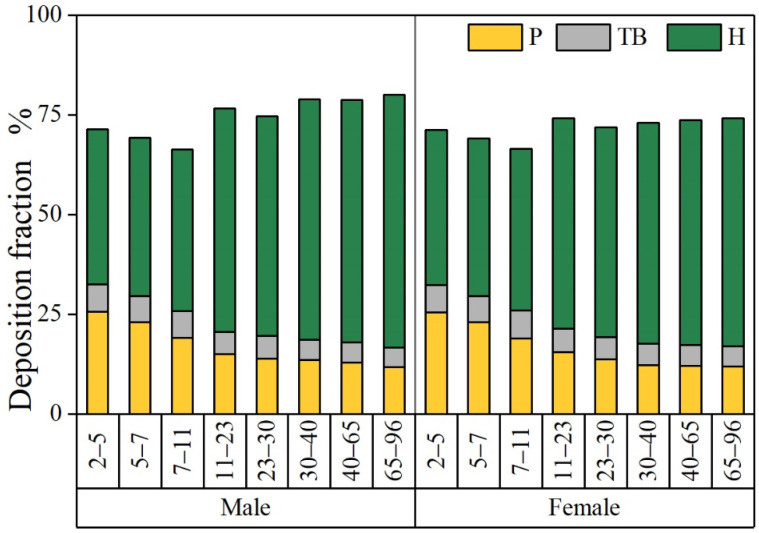
Deposition fractions of PM_2.5_ in different organs of the human body for different population groups. P (yellow portion of bars): pulmonary region of the deposition fraction. TB (the grey portion of bars): tracheobronchial region of the deposition fraction. H (green portion of bars): the upper respiratory tract above the pharynx of the deposition fraction.

**Figure 3 ijerph-19-08915-f003:**
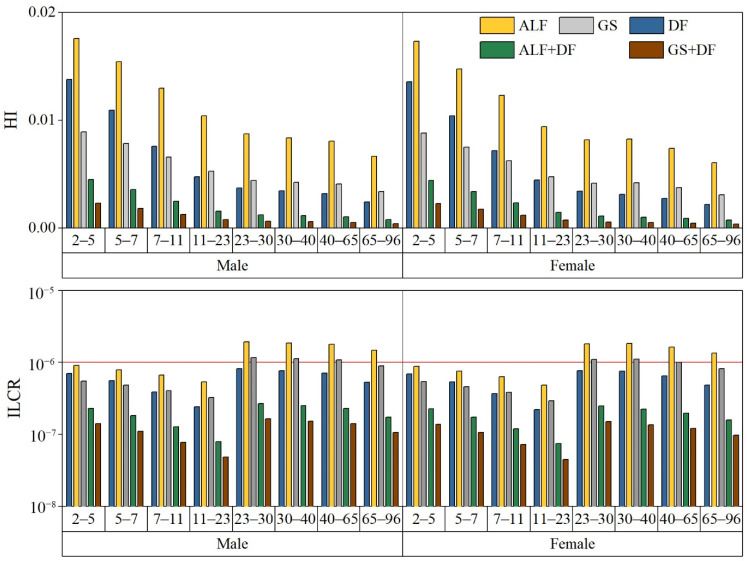
Human health risk assessment of HI and ILCR values of different ages groups for different evaluation concentrations. ALF (yellow portion of bars): artificial lysozyme fluid, represents the result of PM_2.5_ dissolved in acidic simulated lung fluid (pH = 4.5 ± 0.1); GS (the grey portion of bars): Gamble solution, represents the result of PM_2.5_ dissolved in neutral simulated lung fluid (pH = 7.4 ± 0.2). DF (blue portion of bars): the result of PM_2.5_ deposition fraction; ALF + DF (green portion of bars): the result of PM_2.5_ deposition dissolved in the ALF; GS + DF (the brown portion of bars): the result of PM_2.5_ deposition dissolved in GS. The solid red line indicates the value of potential carcinogenic risk.

**Figure 4 ijerph-19-08915-f004:**
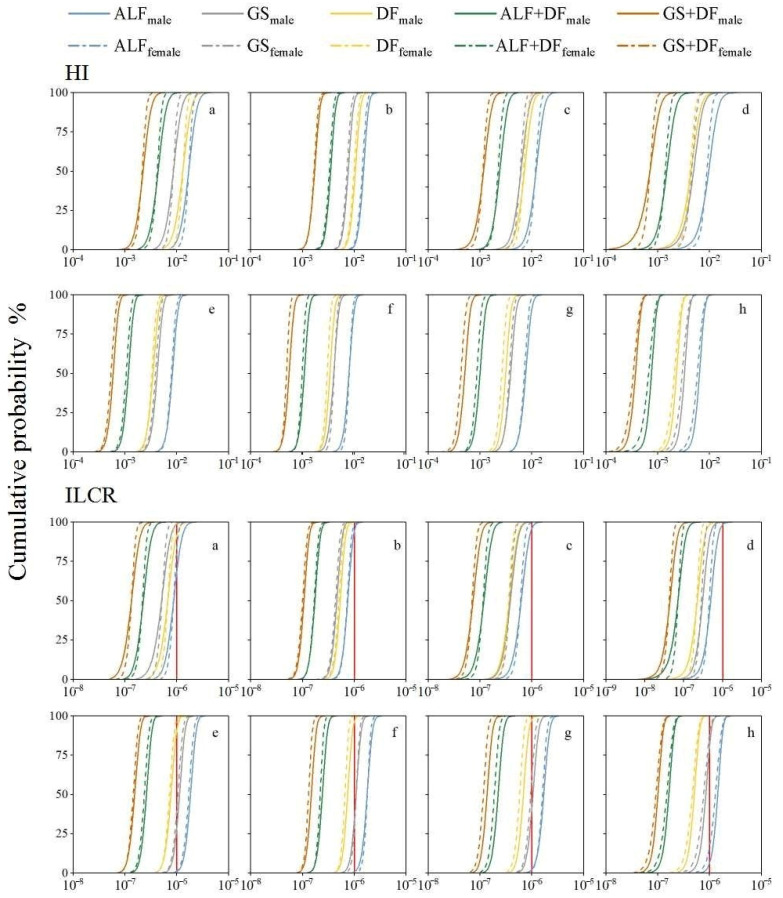
The cumulative probability distribution curve of HI and ILCR for different age groups. ALF: artificial lysozyme fluid, represents the result of acidic simulated lung fluid (pH = 4.5 ± 0.1); GS: Gamble solution, represents the result of neutral simulated lung fluid (pH = 7.4 ± 0.2). DF: represents the result of PM_2.5_ deposition fraction; ALF + DF: the result of PM_2.5_ deposition dissolved in the ALF; GS + DF: the result of PM_2.5_ deposition dissolved in GS. The solid line represents males, and the broken lines represent females. (**a**) represents 2–5-year-old group; (**b**) represents 5–7-year-old group; (**c**) represents 7–11-year-old group; (**d**) represents 11–23-year-old group; (**e**) represents 23–30-year-old group; (**f**) represents 30–40-year-old group; (**g**) represents 40–65-year-old group; (**h**) represents 65–96-year-old group. The solid red line indicates the value of potential carcinogenic risk.

**Table 1 ijerph-19-08915-t001:** PM_2.5_ (μg·m^−3^) and metal (ng·m^−3^) annual mean concentrations and bioaccessibility.

Site	Concentrations					Bioaccessibility	
	PM_2.5_	Metal	Total	ALF	GS	ALF	GS
Xinqu Park	74.7 ± 49.2	Cd	9.99 ± 3.46	3.48 ± 0.97	1.21 ± 0.78	34.8%	12.1%
		Pb	264 ± 109	69.5 ± 30.8	32.8 ± 26.2	26.3%	12.4%
Dongcheng District	88.8 ± 55.4	Cd	11.4 ± 3.30	3.51 ± 1.80	3.04 ± 2.01	30.8%	26.7%
		Pb	452 ± 337	164 ± 99.9	83.1 ± 108	36.2%	18.4%
All study area	81.7 ± 52.9	Cd	10.7 ± 3.46	3.50 ± 1.45	2.13 ± 1.78	32.8%	19.9%
		Pb	358 ± 268	117 ± 87.7	58.0 ± 82.2	32.6%	16.2%

Total represents the total concentration; ALF represents artificial lysosomal fluid; GS represents Gambel solution.

**Table 2 ijerph-19-08915-t002:** Daily inhaled dose (10^−6^ mg·kg^−1^·day^−1^) in PM_2.5_ for different age groups.

Age Groups	Gender	ADD_inh_				LADD_inh_	
		Cd		Pb		Cd	
		ALF	GS	ALF	GS	ALF	GS
2–5	Male	0.427	0.260	14.2	7.07	0.037	0.022
	Female	0.420	0.255	14.0	6.95	0.036	0.022
5–7	Male	0.337	0.205	11.3	5.59	0.029	0.018
	Female	0.322	0.196	10.7	5.33	0.028	0.017
7–11	Male	0.235	0.143	7.82	3.89	0.020	0.012
	Female	0.221	0.135	7.38	3.67	0.019	0.012
11–23	Male	0.147	0.090	4.91	2.44	0.013	0.008
	Female	0.137	0.083	4.58	2.27	0.012	0.007
23–30	Male	0.115	0.070	3.83	1.90	0.043	0.026
	Female	0.105	0.064	3.52	1.75	0.039	0.024
30–40	Male	0.107	0.065	3.57	1.77	0.040	0.024
	Female	0.096	0.058	3.19	1.59	0.036	0.022
40–65	Male	0.099	0.060	3.29	1.64	0.037	0.022
	Female	0.084	0.051	2.82	1.40	0.031	0.019
65–96	Male	0.074	0.045	2.48	1.23	0.028	0.017
	Female	0.068	0.041	2.27	1.13	0.025	0.015

ADD_inh_: the average daily inhaled dose of non-carcinogenic; LADD_inh_: the average daily inhaled dose of metal carcinogens exposed to during daily life; ALF: artificial lysozyme fluid, represents acidic simulated lung fluid (pH = 4.5 ± 0.1); GS: Gamble solution, represents neutral simulated lung fluid (pH = 7.4 ± 0.2).

## Data Availability

Not applicable.

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
