# Peer review of "Inhalation Bioaccessibility and Risk Assessment of Metals in PM2.5 Based on a Multiple-Path Particle Dosimetry Model in the Smelting District of Northeast China"

_ijerph, 2022, doi:10.3390/ijerph19158915_

Round 1

Reviewer 1 Report

The manuscript presents interesting and novel results in the area of ​​public health and risk assessment. I recommend publishing with some adjustments. Although this type of study is for a very restrictive audience.

The manuscript is very well written. The results are well presented and have been discussed properly. The conclusions are supported by the results. The methodology is clear and well defined. Graphs and tables are well presented. A few adjustments needed:

L1: Article;

L29, 31, 153, 208-209 [...]: change μg/m3 to μg.m-3 , check the whole manuscript;

Start the introduction by explaining what PM2.5 is, before dealing with the problem itself;

L111: remove the dot, change "2.3 Chemical analyses." to "2.3 Chemical analyzes";

L277: insert space: change "susceptible[20,21]" to "susceptible [20,21]"; check the entire document, there are excess spaces or lack of spaces.

Reviewer 2 Report

The manuscript number ID: ijerph-1810354 simulates the bioaccessibility of PM2.5 heavy metals in acidic intracellular and neutral extracellular deposition environments by simulating lung fluid; and used the multipath particle dosimetry model to simulate DF of PM2.5; according to the exposure assessment method of U.S. Environmental Protection Agency, the inhalation exposure dose threshold was calculated, and the human health risk with different inhalation exposure doses was compared.

The paper contains some original results and the overall presentation is quite convincing.

In my opinion the following minor corrections are needed before paper publication:

1.     Please ensure the reliability of the study. 2.     Please refer in more detail and in clearly way the advantages of the methodology and summarize possible limitations.

3.     Please reduce the number of self-citations.

Reviewer 3 Report

The study (Inhalation bioaccessibility and risk assessment of metals in PM2.5 based on multiple-path particle dosimetry model in smelting district of Northeast China) includes an investigation of inhalation health risk assessment in order to avoid overestimation. I really liked the paper very much. Very well structured and written. Therefore, I suggest minor revision in order to clarify some points.

Author Response

We have checked the English language and style.
